# Synergistic Modern Global 1 Km Cropland Dataset Derived from Multi-Sets of Land Cover Products

**Chengpeng Zhang [1]** , **Yu Ye [1,2,\*]**, **Xiuqi Fang [1]**, **Hansunbai Li [1]** and **Xueqiong Wei [3]**

1   Faculty of Geographical Science, Beijing Normal University, Beijing 100875, China;
    cpzhang@mail.bnu.edu.cn (C.Z.); xfang@bnu.edu.cn (X.F.); lihansunbai@mail.bnu.edu.cn (H.L.)
2   Key Laboratory of Environment Change and Natural Disaster, Ministry of Education, Beijing Normal
    University, Beijing 100875, China
3   School of Geographical Science, Nanjing University of Information Science & Technology, Nanjing 210044,
    China; xueqiong.wei@nuist.edu.cn
*   Correspondence: yeyuleaffish@bnu.edu.cn

**Abstract:** The quality of global cropland products could affect our understanding of the impacts of cropland reclamation on global changes. With the advancement of remote sensing technology, several global land cover products and synergistic datasets have been developed in recent decades. However, there are still some disagreements among the global cropland datasets. In this paper, we proposed a new synergistic method that integrates the reliability of spatial distribution and cropland fraction on a pixel scale, and developed a modern (around 2000 C.E.) fractional cropland dataset with a 1 km × 1 km spatial resolution on the basis of the spatial consistency of cropland reclamation intensity derived from multi-sets of global land cover products. The main conclusions are shown as follows: (1) The accuracy of spatial distribution assessed by validation samples in this synergistic dataset reaches 87.6%, and the dataset also has a moderate amount of cropland pixels when compared with other products. (2) The reliability of cropland fraction on the pixel scale had been highly improved, and most cropland pixel has a higher fraction (over 90%) in this dataset. The "L" shape of the histogram of pixel numbers with different reclamation intensities is reasonable because it is consistent with the up-scaling results derived from satellite-derived products with high spatial resolutions and the expert knowledge on cultivation. (3) The cropland areas in this non-calibrated result are generally closer to that of FAOSTAT on scales from global to national when compared to other non-calibrated synergistic datasets and original satellite-derived products. (4) The reliability of the synergistic result developed by this method might be decreased to some degree in the regions with high discrepancies among the original multi-sets of cropland datasets.

**Keywords:** cropland cover; dataset; overlay; consistency; synergistic; 1 km × 1 km

## 1. Introduction

Nearly two-fifth of land surface has been transformed into agricultural land to satisfy basic living needs (e.g., food, timber, and raw textile materials) for human beings so far [1–4]. As one of the drastic and long-standing land-use types, cropland has possessed nearly 12.2% of the global land area since the Holocene [4]. Besides reshaping the landscape directly, the process of cropland reclamation and land cover change is also an important anthropogenic driving force of global change [5]. Due to the changes in the physical conditions (e.g., radiation forcing, albedo, roughness, and evapotranspiration) of the original land surface [6,7] and the processes of biogeochemical cycles (e.g., the carbon cycle and nitrogen cycle) [8–11], croplands have profound effects on global and regional terrestrial ecosystems [12,13], water cycles [14], and climate change [15]. Therefore, accurately assessing the impact of croplands on

global change highly depends on the reliability of global cropland products [16–18]. The reliability of these datasets is mainly reflected in three aspects of the accuracy of spatial distribution, the rationality of reclamation intensity (cropland fraction) per pixel, and the moderate total cropland amount [19,20].

With the continuous advances in remote sensing technology [21,22], a series of modern global land cover products have gradually developed in recent decades to serve the above-mentioned studies [10,23,24], such as IGBP-DISCover and the GFSAD-Cropland products developed by the U.S. Geological Survey [25,26], GLC-MODIS developed by Boston University [27], CCI_LC (climate change initiation land cover) product developed by the European Space Agency [28], and FROM-GLC developed by Tsinghua University [29], etc. The operation time of these satellite-derived products mainly began during the 1990s [23]. Due to the improvement in the spatial resolution of the original satellite images, the development of the interpretation method and the acceleration of computational efficiency, the spatial resolution of these products has gradually been as fine as 30″ × 30″(approximately 1 km × 1 km at the equator) from $1° × 1°$ (approximately 100 km × 100 km at the equator) earlier [23,30,31]. In recent years, some institutions have developed new products with an even finer resolution of 10 m × 10 m [32]. In addition, the interpretation accuracies of these products have also improved significantly. For example, the overall classification accuracy of IGBP-DISCover (1992–1993) released in 2000 is only 66.9% [23]. While the accuracy of the GlobeLand30 product (2010) released in 2015 has increased to 80.3% [23]. However, uncertainties in each satellite-derived dataset and obvious differences among the datasets still existed, such as the spatial distribution, the areal amount [33–39] mainly caused by the discrepancies in various satellite images they used, different interpretation methods, and land cover classification systems (LCCSs), adopted by different products [35].

To improve the spatial distribution accuracy of the existing products and to reduce the discrepancies in the multi-sets of products caused by different LCCSs, some fractional global/regional land cover products or cropland datasets with moderate resolutions (1 km × 1 km) have been developed by synergizing the multi-sets of satellite-derived products [40–43]. Among these synergized datasets, some are for croplands [18,44–46]. The cropland synergistic methods made full use of the spatial distribution information in the original products. Generally, either the mean value of the estimated cropland fraction from all products [47] or the cropland fraction from a single optimal product [18] is defined as the synergistic result on pixel scale for the assignment of cropland fraction in the synergistic process. The total cropland amount in some synergistic products has been further calibrated with national or subnational statistical data [41,44,48,49]. Compared with the satellite-derived Boolean products, the synergized fractional cropland datasets have the advantages that the accuracy improvement of spatial distribution [23] and the anthropogenic land-use intensity has been expressed much accurately with fractional format [18].

Although the quality of the synergistic cropland products has greatly improved in terms of having a more reasonable, and an appropriate amount of spatial distribution (only for statistics-calibrated results), some problems still exist. First, using the mean value of several products as the synergized dataset's cropland fraction causes the pixels with different reclamation intensities (over a wide proportional range) in the original products to be converted to the same fraction. Second, calibrating the total cropland amounts of the original synergized dataset without considering the systematic difference between the satellite-derived products and statistics would narrow the cropland spatial distribution or reduces the reclamation intensity in the dataset. The cropland area summed from the spatial-based products is usually more than that from the statistical data. For example, some studies show that the cropland areas in most satellite-derived products are over 40% larger than those from the statistics in China [50,51].

The purpose of this study is to develop a fractional synergized cropland dataset with the improvement on the reliability of spatial distribution and cropland fraction on the pixel scale, by synthetically considering the three aspects of spatial distribution, cropland fraction per pixel, and total amount. Nine of the most widely used global land cover products/cropland datasets, including two satellite-derived global land cover products with high resolution, were collected to extract the cropland

subsets. A modern global 1 km fractional cropland dataset has been developed based on the spatial consistency of the cropland reclamation intensity among these datasets.

## 2. Materials and Methods

### 2.1. Data Sources

Nine global land cover products/synergized datasets with 1 km × 1 km or finer spatial resolutions are used in this study (Table 1). The years represented by these products range from 1992 to 2005. The time of all these datasets is around 2000 C.E. Six of the datasets are original satellite-derived Boolean type products, including GLC-MODIS [27], GLC-UMD [52], GLC2000 [53], GLCNMO [54], ESA-CCI-LC [28], and GlobeLand30 [55]. The other three datasets are in fractional format, including GLC-Consensus [47], HybridCropland [18], and GLC-Share [41]. The total cropland amounts in the last two products were further calibrated with FAOSTAT statistical data.

**Table 1.** Detailed information on cropland classes defined in nine global land cover/cropland products.

| No | Product | Type | Resolution | Year | LCCS | Cropland Classes/Reclamation Ratio |
|----|---------|------|-----------|------|------|-----------------------------------|
| 1 | GLC-UMD | Boolean | 1 km | 1992–1993 | Modified IGBP | 11. Croplands (81–100%)<br>Other classes (0–80%) |
| 2 | GLC-MODIS | Boolean | 1 km | 2001 | IGBP | 12. Croplands (61–100%)<br>14. Cropland/Natural Vegetation Mosaics (11–60%)<br>Other classes (0–10%) |
| 3 | GLC2000 | Boolean | 1 km | 2000 | FAO | 16. Cultivated and managed areas (61–100%)<br>17. Mosaic: Cropland/Tree Cover/Other natural vegetation (16–60%)<br>18. Mosaic: Cropland/Shrub and/or grass cover (16–60%)<br>Other classes (0–15%) |
| 4 | GLCNMO | Boolean | 500 m | 2003 | Modified FAO | 11. Cropland (61–100%)<br>12. Paddy field (61–100%)<br>13. Cropland/other vegetation mosaic (16–60%)<br>Other classes (0–15%) |
| 5 | ESA-CCI-LC | Boolean | 300 m | 2000 | Modified FAO | 10. Cropland, rainfed (71–100%)<br>11. Herbaceous cover (71–100%)<br>12. Tree or shrub cover (71–100%)<br>20. Cropland, irrigated or post-flooding (71–100%)<br>30. Mosaic cropland/natural vegetation (51–70%)<br>40. Mosaic natural vegetation / cropland (21–50%)<br>Other classes (0–20%) |
| 6 | GlobeLand30 | Boolean | 30 m | 2000 | China | 10. Cropland (100%) |
| 7 | HybridCropland | Fraction | 1 km | around 2000 | —— | (0–100%) |
| 8 | GLC-Share | Fraction | 1 km | around 2000 | Modified FAO | 2. Cropland (0–100%) |
| 9 | GLC-Consensus | Fraction | 1 km | around 2000 | Modified IGBP&FAO | 7. Cultivated and managed vegetation (0–100%) |

Detailed information on the croplands defined in the nine datasets is shown in Table 1, including the data set name, spatial resolution, years represented by the satellite image, LCCS type, cropland-related class name/code in each product and the specific description. According to the differences in LCCSs, the six Boolean-type products can be classified into three categories. Among them, GLC-MODIS and GLC-UMD adopt the modified IGBP-LCCSs; GlobeLand30 uses China's independent classification system (the pasture and fruit trees are also identified as cropland); the remaining three mainly take the FAO-LCCSs or its modified systems. In addition, even in the datasets with similar LCCSs, the cropland definition still has some discrepancies in the specific crop types included in croplands, the difference of land-use methods, and the range of crop proportion in different Boolean classes of croplands [34].

To assess the spatial distribution accuracy of the synergistic dataset developed by this study, the independent validation samples of the FROM-GLC dataset released by Tsinghua University (link: http://data.ess.tsinghua.edu.cn/) had also been collected. This dataset contains approximately

3000 cropland validation samples covering both the main agricultural areas and the sparse cultivation regions around the world.

To further assess the cropland areal amount of the synergistic dataset developed by this study on continental and national scales, the statistical data released by FAOSTAT (link: http://www.fao. org/faostat/) were also collected. FAOSTAT is easily accessed and the most authoritative available statistical data on a global scale. Its clear and specific cropland definition is suitable for most studies on considering the total amount of cropland. To match the cropland definition of the satellite-derived datasets, the FAOSTAT arable land and permanent crops area for each country in 2000 C.E. was selected.

## 2.2. Methods

### 2.2.1. Cropland Subset Extraction and Reclamation Intensity Reclassification

The cropland dataset was extracted from each original dataset in order to obtain a unified reclassified result for cropland reclamation intensity and to further conduct consistency overlay analysis. First, the World Geodetic System 1984 (WGS84) was chosen as the standard geographic coordinate system for all original products. Then, the spatial resolution of all the datasets (except GlobeLand30 and ESA-CCI-LC) were resampled to 30″ × 30″ (~1 km × 1 km at the equator) by using the nearest neighbor method in ArcGIS. Next, every land cover class in the four Boolean datasets (GLC-UMD, GLC-MODIS, GLC2000, and GLCNMO) was converted into its corresponding cropland reclamation intensity intervals according to the cropland proportional range in each class.

According to the cropland intensity proportional ranges defined in different classification systems, the entire fractional range (0–100%) is divided into 4 levels. That interval of 0–15% is level 1; 16–60% is level 2; 61–80% is level 3; and 81–100% is level 4 (Figure 1). If a cropland class with a wide range contains more than one level, all corresponding level codes should be assigned to it. For example, the cropland proportion for class 12 in the GLC-UMD is 81–100% that is marked with a level code of 4; while the other classes with a cropland proportion less than 80% in this product are labeled with 1, 2, and 3. For GLC-MODIS, class 11 is marked with 3 and 4 according to its proportion ranges from 61% to 100%; and class 14 is marked with 2 while the other classes are marked with 1. For the three synergized fractional datasets, we only need to set the corresponding code for each level after the reclassification process according to the fractionally divided criteria.

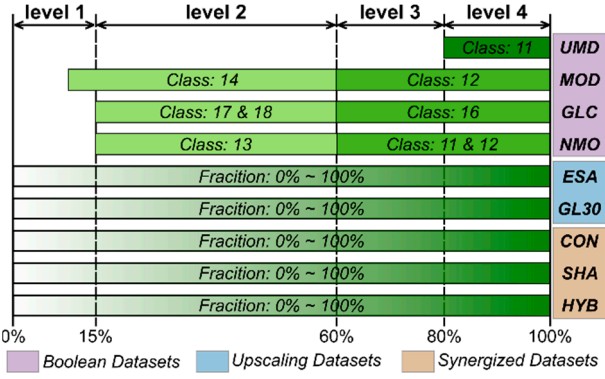

**Figure 1.** A schematic diagram of the cropland reclamation intensity reclassification. (GLC-UMD is abbreviated as UMD; GLC-MODIS is abbreviated as MOD; GLC2000 is abbreviated as GLC; GLCNMO is abbreviated as NMO; ESA-CCI-LC is abbreviated as ESA; GlobeLand30 is abbreviated as GL30; GLC-Consensus is abbreviated as CON; HybridCropland is abbreviated as HYB.).

For GlobeLand30, all 853 original tile images were merged into two hemispheres. To facilitate the cropland area calculation and reduce the raw data storage, we chose the Albers Equal Area Conic projection as the projected coordinate system in the mosaic process and set the spatial resolution as 50 m × 50 m. Next, 1 km × 1 km grids of global potential cropland distribution were made by overlaying all cropland classes and cropland/natural vegetation mosaic classes derived from the

Boolean products with coarse resolutions in ArcGIS. Then, the zonal statistical function in the ArcGIS spatial analysis tool was adopted to calculate the cropland proportion in each 1 km × 1 km grid for obtaining the up-scaling fractional dataset. On 50 m spatial resolution, each Boolean pixel can represent approximately 100% of each land cover type. Therefore, the cropland fraction in each 1 km × 1 km grid cell is the cropland pixel numbers divided by the total pixel numbers in the grid. While in the ESA-CCI-LC, the croplands cannot occupy the whole pixel under that spatial resolution (300 m × 300 m). Thus, the proportion of all cropland classes in its LCCS should be redefined at first. Like most relevant studies [47], this study also adopted the average value of the corresponding fractional range defined in each class as the cropland fraction on pixel scale. Then, a similar up-scaling method as GlobeLand30's was used to generate the 1 km fractional result. Finally, the two new fractional datasets were reclassified into 4 levels and marked with corresponding level codes. Instead of resampling directly, converting these products into fractional results based on an up-scaling method could partially solve the problem where it is impossible to estimate the exact cropland fraction in Boolean products with a coarse resolution.

### 2.2.2. Analysis of Spatial Consistency of Cropland Reclamation Intensity and Cropland Dataset Synergizing

The flow chart of the synergistic procedure is shown in Figure 2. The analysis of spatial consistency of the cropland reclamation intensity levels among nine cropland datasets was carried out. Based on the overlay result of intensity levels, two temporary process datasets were generated including cropland reclamation intensity levels (the most likely cropland fractional ranges per pixel) and the consistency grade (the maximum number represents how many datasets have the same intensity levels). Then, we set the prioritized order for several original cropland datasets in each continent by comprehensively considering the spatial resolution of the original products, the interpretation accuracy and the authority of regional experts/institutions, etc. Finally, the optimal product's cropland fraction was assigned to synergize the cropland pixels. The detailed steps are described as follow.

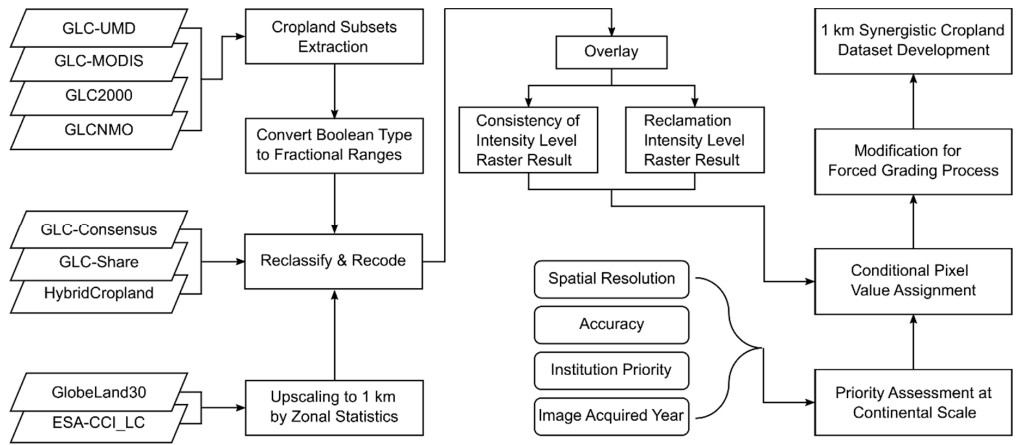

**Figure 2.** The flow chart of the global 1 km cropland synergistic method based on the cropland intensity consistency analysis of multi-sets of land cover products.

In each original cropland dataset's reclassification result, the number of reclamation intensity levels ranges from 2 to 4 (see Supplementary materials S2). There are 55,296 cases following to overlay nine datasets. The encoding rules were set up by using an individual ID code to represent each case. The code can reflect both the intensity levels and the original cropland dataset. The purpose of the encoding process is to count the numbers of the same intensity levels indicated by these datasets and select the intensity levels with a unique mode as the fractional ranges of the synergistic cropland pixels. When there are five or more datasets indicating that the reclamation intensity is at the same level, the cropland fractional range of the synergistic pixel must be in this level. When the number in an intensity level is 4 and the numbers of the other levels are less than 4, the fractional range of the

synergistic pixels must be that level. However, if a certain intensity level has more than one majority, the intensity level would be determined by further taking the original dataset's spatial resolution, interpretation accuracy, and other referential factors into consideration. For example, the cropland intensity of a pixel is indicated at level 1 by only 1 dataset, at level 2 by 2 datasets, and at level 3 and level 4 by 3 datasets, separately; but when the pixel values of the cropland fraction in the two most trusted datasets of GlobeLand30 and ESA-CCI-LC are both at level 4, then the cropland fractional range of the combination pixel should be identified as level 4. The cropland intensity levels of all cases are given in the Supplementary materials S2.

Based on two temporary process raster datasets generated from the above mentioned spatial analysis and the pre-set prioritized datasets in each continent, we further carried out the 1 km cropland dataset synergizing process for each continent. Considering the similarities across different continents and the synergistic workload, the six continents were grouped as follows when the prioritized dataset in each continent was determined. That is, Asia and Oceania in a group, Europe in a group, and the Americas and Africa in a group. The assignment operation is conducted by using map algebra in the ArcGIS spatial analysis toolbox to execute the nested conditional sentences. Figure 3 shows the principle behind the pixel value assignment process. The more specific assignments with conditional sentences are described in the Supplementary materials S3.

### 2.2.3. Modification of the Primary Synergistic Result

There is an unavoidable disadvantage in the forced reclassifying process of cropland reclamation intensity. If one pixel value in a dataset is 81% and only 79% in another dataset in the same location, the two values would be reclassified into two different levels according to the given reclassification rules. Therefore, the following method was adopted to modify the primary result. That is if the difference between the first priority dataset and second priority dataset is within ±10%, the synergistic pixel values are assigned with the cropland fraction of GlobeLand30 (it is regarded more reliable because its cropland fraction is generated from a dataset with the highest spatial resolution). These pixels in primary synergistic result were replaced by GlobeLand30's fractions.

### 2.2.4. Accuracy Assessment of the Cropland Datasets

Approximately 3,000 cropland validation samples from the original data provided by FROM-GLC were extracted. Next, a 500 m circular buffer for each point was built in ArcGIS and made an external square polygon for each circle. A total of 2784 grids (1 km × 1 km) were generated. We used the zonal statistical tool to calculate how many pixel numbers were identified as croplands in each cropland dataset under the locations of these samples (for the Boolean dataset, the cropland class and cropland/natural vegetation class are counted as cropland pixels; for the proportional dataset, a cropland fraction >10% is counted as a cropland pixel). The cropland pixel numbers divided by the total validation numbers (2784) had been considered as an indicator to assess the spatial distribution accuracy of the cropland dataset.

$$Accuracy_{Boolean} = Pixel_{cropland} / 2784 \qquad (1)$$

where $Accuracy_{Boolean}$ is the accuracy of the Boolean dataset and $Pixel_{Cropland}$ is the pixel numbers identified as the cropland type.

$$Accuracy_{Fraction} = Pixels_{ratio>10} / 2784 \qquad (2)$$

where $Accuracy_{Fraction}$ is the accuracy of the fractional dataset and $Pixel_{Cropland}$ is the pixel numbers with a cropland fraction >10%.

### 2.2.5. The Quantitative Assessment of Cropland Area of Fractional Cropland Datasets

The cropland areas of all continents (except the Antarctic) and the world were added up from national cropland areas in the FAOSTAT. Considering that cropland areas can only be calculated from

the dataset with exact cropland fraction on pixel scale, the cropland areas in different fractional datasets were quantitatively assessed by this reference statistical data. The relative differences of cropland area between fractional dataset and FAOSTAT were calculated as the below formula:

$$\text{Diff} = (Area_{Frac} - Area_{FAO})/Area_{FAO} \tag{3}$$

where *Diff* is the relative difference of cropland area between fractional dataset and the FAOSTAT in an evaluated spatial unit; $Area_{Frac}$ is the cropland area calculated from one fractional dataset in this evaluated spatial unit; $Area_{FAO}$ is the cropland area indicated by FAOSTAT in this evaluated spatial unit.

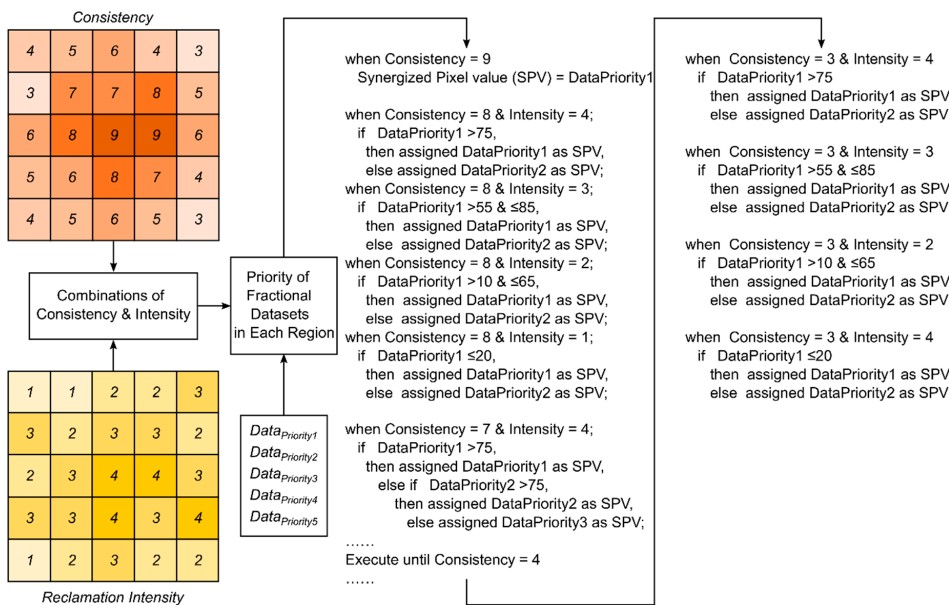

**Figure 3.** Schematic flowchart of the synergistic cropland pixel assignment based on the analysis of spatial consistency of cropland reclamation intensity. Consistency raster data of the intensity levels (top left) have 7 values, including 9, 8, 7, 6, 5, 4, and 3. The values indicate how many datasets have the same intensity levels. The cropland intensity results (bottom left) is divided into 4 levels: 0–15% is level 1, 16–60% is level 2, 61–80% is level 3, and 81–100% is level 4. Combined with a pre-set data prioritization order, the conditional sentences of the pixel assignment method are shown in the right panel. When the consistency raster value is 9, the synergistic pixel value is assigned with the first priority dataset's cropland fraction; when the consistency raster value is 8 and the cropland intensity level raster value is 4, if the cropland fraction in the first priority dataset is ≥75% (considering the inevitable defect in the process of forced reclamation rate reclassification, where ±5% tolerance is set), then this cropland fraction is assigned to the synergistic pixel; otherwise, the synergistic pixel value must be the fraction of the second priority dataset. The rest of the combination can be deduced by this assignment principle. When the consistency raster value is only 3, it indicates that a huge discrepancy in cropland intensity existed among these datasets. Therefore, we assign the pixel value for the synergistic cropland dataset by only depending on the cropland fraction of the first priority or second priority datasets. If the fraction in the first priority dataset is within the corresponding intensity levels, then this value is assigned to the synergized pixels; otherwise, the value of the second priority dataset is assigned.

## 3. Results and Interpretations

### 3.1. Spatial Distribution of the Synergistic Cropland Dataset

The global 1 km fractional synergistic cropland dataset developed in this study is shown in Figure 4. The spatial distribution of the croplands show a more concentrated distribution with more higher intensity (the cropland proportion per pixel is >60%) in the world's main agricultural regions,

including the Indian Plain, the Gangetic Plain, the North China Plain in Asia, the Eastern European Plain, the Great Plains in North America, the La Plata Plain in South America, and the Chad Basin in Africa. In addition to these large regions, small amounts of intensive cropland are concentrated in some irrigated agricultural regions such as the Nile Valley in Egypt and the oasis in Central Asia. Croplands with moderate reclamation intensity (the cropland fraction per pixel is less than 60% but more than 20%) are mainly distributed around the abovementioned high-intensity regions, such as the Agro-Pastoral ecotone in China, the southwestern part of the Brazilian Plateau and the Guinea Plateau in Africa. Croplands with low reclamation intensity (the cropland proportion per pixel < 20%) show a sporadic distribution around the world, such as the vast areas in sub-Saharan Africa, the northern part of South America, the Northern Europe and the world-wide hilly mountainous regions. Table 2 also shows the accuracy of cropland spatial distribution evaluated by the FROM-GLC validation samples in nine global cropland datasets.

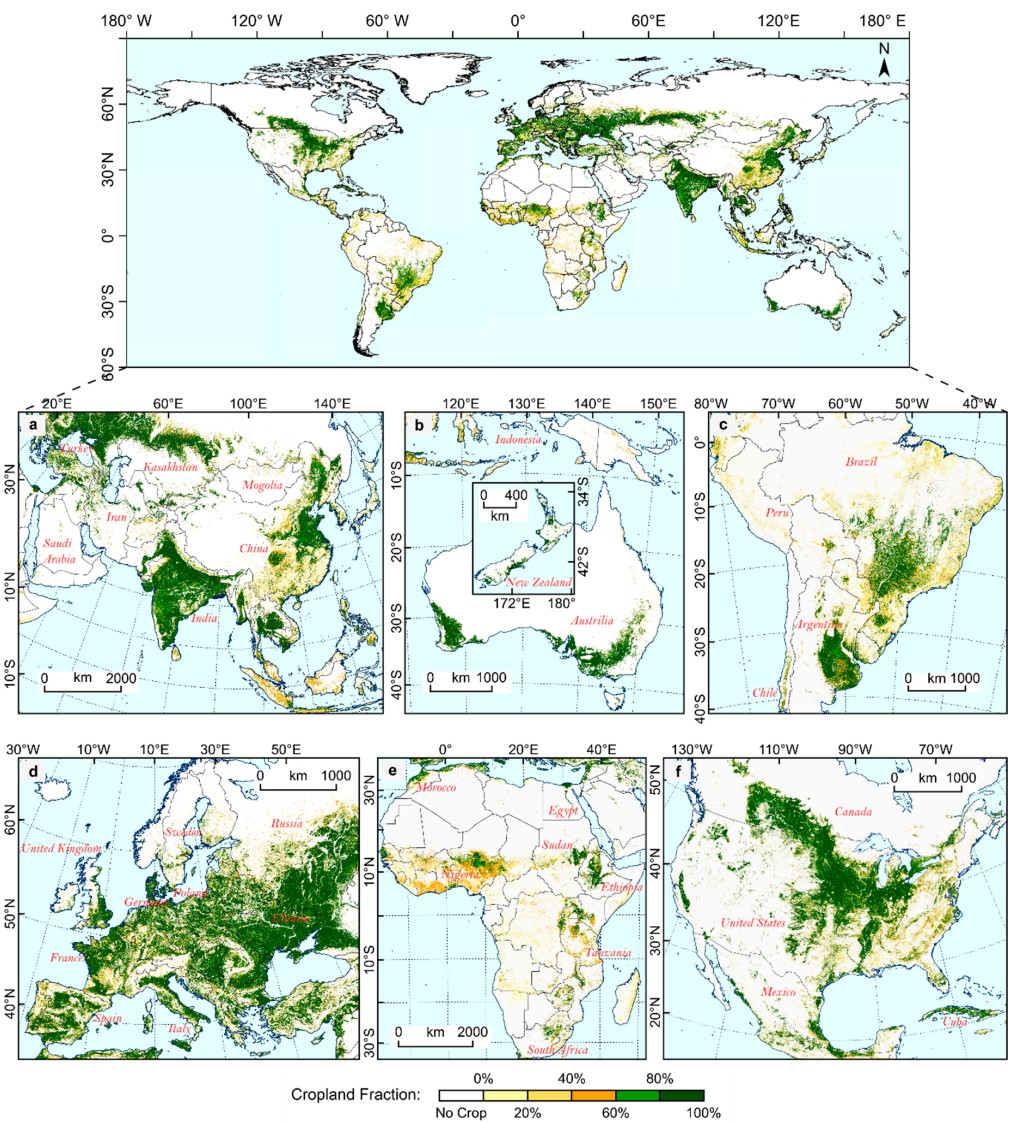

**Figure 4.** Spatial distribution maps of the global 1 km synergistic cropland dataset developed in this study. The results are displayed on a continental scale, where (**a**) is Asia, (**b**) is Oceania, (**c**) is South America, (**d**) is Europe, (**e**) is Africa, and (**f**) is North America and Central America. The cropland fractions (1–100%) was reclassified into five classes by every 20% interval: higher intensity (61–100%), moderate intensity (20–60%) and lower intensity (1–20%) (The original synergistic cropland dataset is available in Supplementary materials).

**Table 2.** The accuracy of cropland spatial distribution evaluated by the FROM-GLC validation samples in nine global cropland datasets.

| Product | Accuracy | Product | Accuracy |
|---------|----------|---------|----------|
| GlobeLand30 | 89.94% | GLC-Consensus | 95.47% |
| ESA-CCI-LC | 87.68% | HybridCropland | 87.61% |
| GLC-MODIS | 67.53% | This Study | 87.61% |
| GLC2000 | 66.31% | GLC-Share | 87.07% |
| GLC-NMO | 64.19% | | |
| GLC-UMD | 40.01% | | |

Note: The results of the satellite-derived Boolean products are shown in the left column, the synergistic datasets are shown in the right column, and the spatial distribution accuracies of the two formatted datasets are sorted in descending order.

### 3.2. Accuracy of the Reclamation Intensity Characteristics in the Synergistic Cropland Dataset

We calculated the pixel amount in each cropland reclamation intensity level from the synergistic dataset and compared it with the other five fractional datasets. The results are shown in Figure 5. The total numbers of cropland pixels in the synergistic dataset exceeds $40 \times 10^6$. That is very similar to the other four fractional datasets, except for the GLC-Consensus.

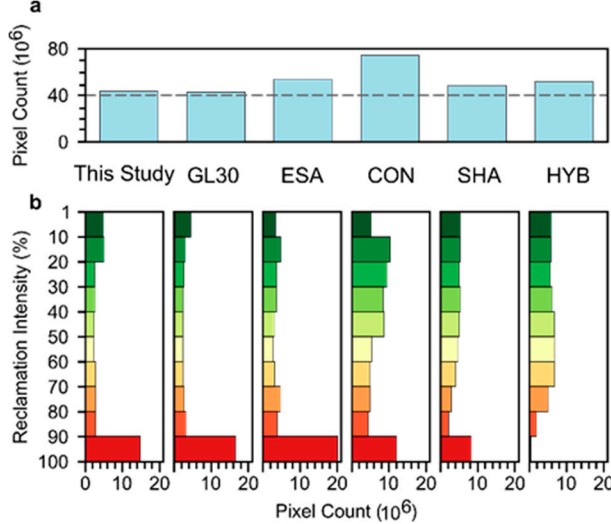

**Figure 5.** Pixel counts with different reclamation intensities in six fractional datasets. (**a**) The total numbers of cropland pixels (the cropland fraction per pixel is >1%) for each dataset; (**b**) the numbers of cropland pixels within every intensity level (the whole 100% interval was divided into 10 equal intervals).

However, obvious differences were shown in the histogram's quantitative distribution of different levels among the fractional datasets. The histograms of the intensity levels in the up-scaling datasets of GlobeLand30 and ESA-CCI-LC displayed an "L" shaped curve (the y-axis represents the intensity levels in order from lowest to highest). It means that most cropland pixels are always with high fractions. Namely, the proportion of cropland pixels with over 90% fractions is much more than the amount of pixel with fractions less than 90%. The histogram of our synergized cropland data shows highly consistent characteristics with the two up-scaling datasets. The characteristics of the histogram from the other three fractional datasets display entirely different appearances. In these three datasets, the pixel numbers at the highest intensity level possess a significantly small percentage of the total amount of cropland pixels, while the pixel amounts at the moderate intensity levels are relatively higher. For example, the histogram of HybridCropland shows a completely different shape from the histograms of the two up-scaling results and our dataset.

### 3.3. Spatial Accuracy and Areal Reasonability of the Synergistic Cropland Datasets

In the absence of statistical calibration, the cropland area in our synergistic dataset should be assessed to detect if it has a large error in the amount on different scales. The global cropland area in our synergistic dataset is only 12% more than the statistical cropland area in the FAOSTAT. It is much less than the over 50% relative differences in cropland area in ESA-CCI-LC and GLC-Consensus.

On a continental scale, the relative differences of cropland area to the FAO statistics in our synergistic dataset are also better than the other three datasets (Table 3). In most continents, the relative differences are all within 15%, although it exceeds 39% in Latin America. In contrast, the cropland areas of GLC-Consensus on most continents (except North America) are far greater than the FAO statistics with the relative differences all exceeding 50%. The ESA-CCI-LC cropland areas in Latin America and Africa are substantially larger than the FAO cropland areas with relative differences of 165% and 157%, respectively, while the area in North America is 35% less than the FAO statistics. In GlobeLand30, the relative differences in cropland area in Asia, Oceania, Africa, and North America are within ± 20%, and this difference is approximately 50% in Latin America.

**Table 3.** The continental cropland areas calculated from four non-statistical calibrated cropland datasets and their relative differences compared with FAOSTAT data.

| Region | This Study | | ESA-CCI-LC | | GlobeLand30 | | GLC-Consensus | | FAO |
|--------|------|------|------|------|------|------|------|------|------|
| | **Area** | **Diff** | **Area** | **Diff** | **Area** | **Diff** | **Area** | **Diff** | **Area** |
| AS | 6.43 | ▼12% | 7.87 | 37% | 6.88 | 19% | 9.06 | ▲57% | 5.76 |
| EU | 4.00 | 28% | 3.92 | ▼25% | 4.21 | 35% | 4.79 | ▲53% | 3.13 |
| OA | 0.54 | 7% | 0.48 | ▼−5% | 0.58 | 13% | 0.83 | ▲63% | 0.51 |
| AF | 2.15 | ▼−8% | 6.04 | ▲157% | 2.03 | −14% | 4.51 | 92% | 2.35 |
| NA | 2.33 | −4% | 1.56 | ▲−35% | 2.48 | ▼3% | 2.60 | 8% | 2.41 |
| LA | 2.27 | ▼39% | 4.33 | ▲165% | 2.42 | 48% | 4.07 | 149% | 1.63 |
| World | 17.73 | ▼12% | 24.20 | 53% | 18.59 | 18% | 25.85 | ▲64% | 15.79 |

Note: cropland area is in $10^6$ km²; relative difference = $(AREA_{Frac}-AREA_{FAO})/AREA_{FAO}$; unit: %; the numbers marked with▼ indicate that the relative difference in the continental cropland area among the four datasets is the smallest; numbers marked with ▲ indicate that the relative difference in the continental cropland area among the four datasets is the largest.

The national cropland area derived from the six fractional cropland datasets and its relative differences with the FAOSTAT data is shown in Figure 6. The scatter points near the 1:1 line indicate smaller relative differences. The more uniformity the point sizes have, the more significant systematic difference exists between the FAO statistical data and the satellite-derived fractional data. In the synergistic dataset developed in this study, the scatter points are closer to the 1:1 line in the countries with greater cropland areas, but more scattered in the countries with smaller cropland areas (Figure 6f). Since the GLC-Share and HybridCropland data were calibrated with the statistical data, the national cropland areas calculated from the two datasets are more consistent with the FAOSTAT areas (Figure 6a,b, respectively). Most countries in the GLC-Consensus and ESA-CCI-LC datasets show extremely large cropland areas compared to the FAO statistical areas that are the relative differences are greater than 200% even for some countries where cropland areas are greater than $10^4$ km² (Figure 6c,d). The relative differences in cropland area in GlobeLand30 are relatively smaller, but some countries' cropland areas are underestimated indicated by some scatter points below the 1:1 line (Figure 6e). The scatter points of the synergistic dataset developed in this study are similar to the GLC-Share and HybridCropland data in the countries with greater cropland areas, and slightly closer to the 1:1 line than the GlobeLand30 dataset in the countries with smaller cropland areas. It means decreased relative differences and less underestimation in national cropland area (Figure 6f).

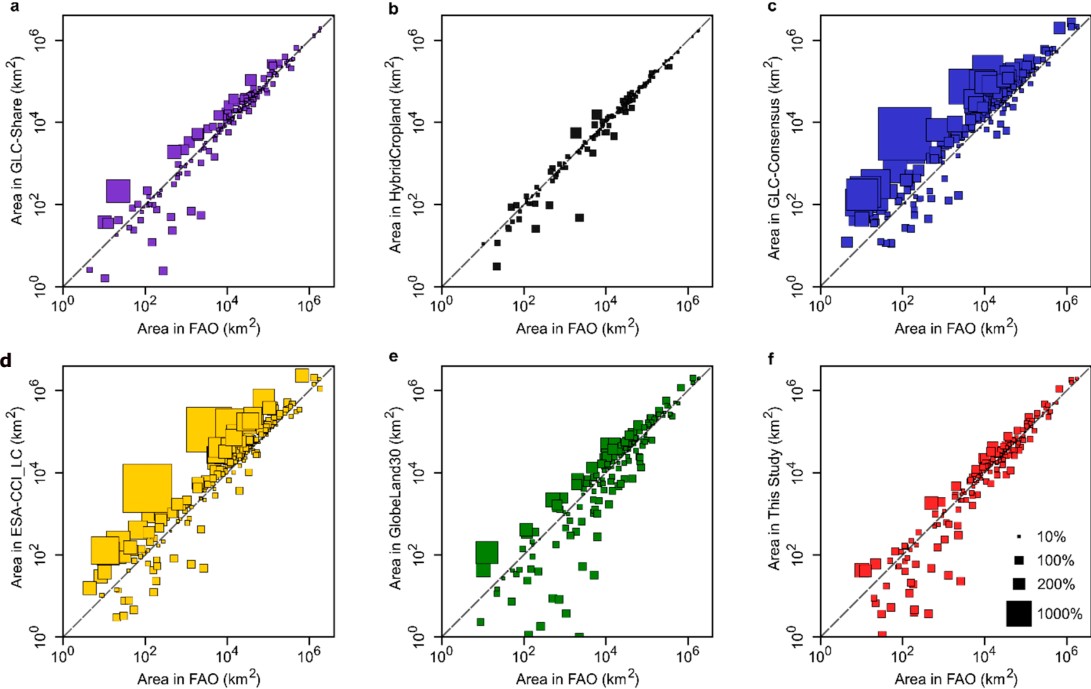

**Figure 6.** Relative differences in the national cropland areas between six fractional datasets and FAOSTAT data. The x-axis represents the national cropland area in FAOSTAT (2000 C.E.); the y-axis represents the national cropland area in the fractional dataset; the point size indicates the relative difference ($|AREA_{Frac} - AREA_{FAO}|$)/$AREA_{FAO}$) in cropland area between the fractional dataset and FAOSTAT dataset, where (**a**) shows the GLC-Share results, (**b**) shows the HybridCropland results, (**c**) shows the GLC-Consensus results, (**d**) shows the ESA-CCI-LC results, (**e**) shows the GlobeLand30 results and (**f**) shows the synergistic cropland dataset results developed in this study.

We further sorted the national cropland areas of FAOSTAT in descending order and selected the top 30 countries. The relative differences in the four non-calibrated datasets are shown in Figure 7. There are only three countries with relative differences exceeding ±50%, which are located in Africa and Latin America in our dataset. In GlobeLand30, the number of countries with differences over ±50% is eight, and nine countries have fewer cropland areas than the FAO's, of which five in Asia and Africa are most serious (more than −20%). The national cropland areas obtained from ESA-CCI-LC and GLC-Consensus are generally larger than the FAO areas. The differences in approximately half of the countries exceed 50% in ESA-CCI-LC, and 10 countries exceed 50% (over 100% in six countries) in GLC-Consensus's.

From a national perspective, the countries with substantial cropland amount indicated by satellite-derived datasets are China, Thailand, and Myanmar in Asia; France and Germany in Europe; Sudan, Ethiopia, and Tanzania in Africa and Brazil and Argentina in South America, etc. Generally, most of the cropland in these above-mentioned countries are distributed in regions with favorable agricultural conditions and also have a relatively higher reclamation intensity (Figure 4). Although China has the third-largest cropland area in the world, the cropland areas in these datasets are always extremely large when compared to that of the FAOSTAT (it exceeds 100% in GLC-Consensus), the closest result to the statistics was obtained in this study, but still 38% more than the FAOSTAT. The countries with obviously fewer cropland amounts as indicated by the satellite-derived datasets are Pakistan in Asia, Finland in Europe, Niger in Africa, and Canada in North America. In addition, cropland areas in some countries, such as Pakistan and Iran, in our synergistic dataset are severely less than the FAOSTAT when comparing with other datasets.

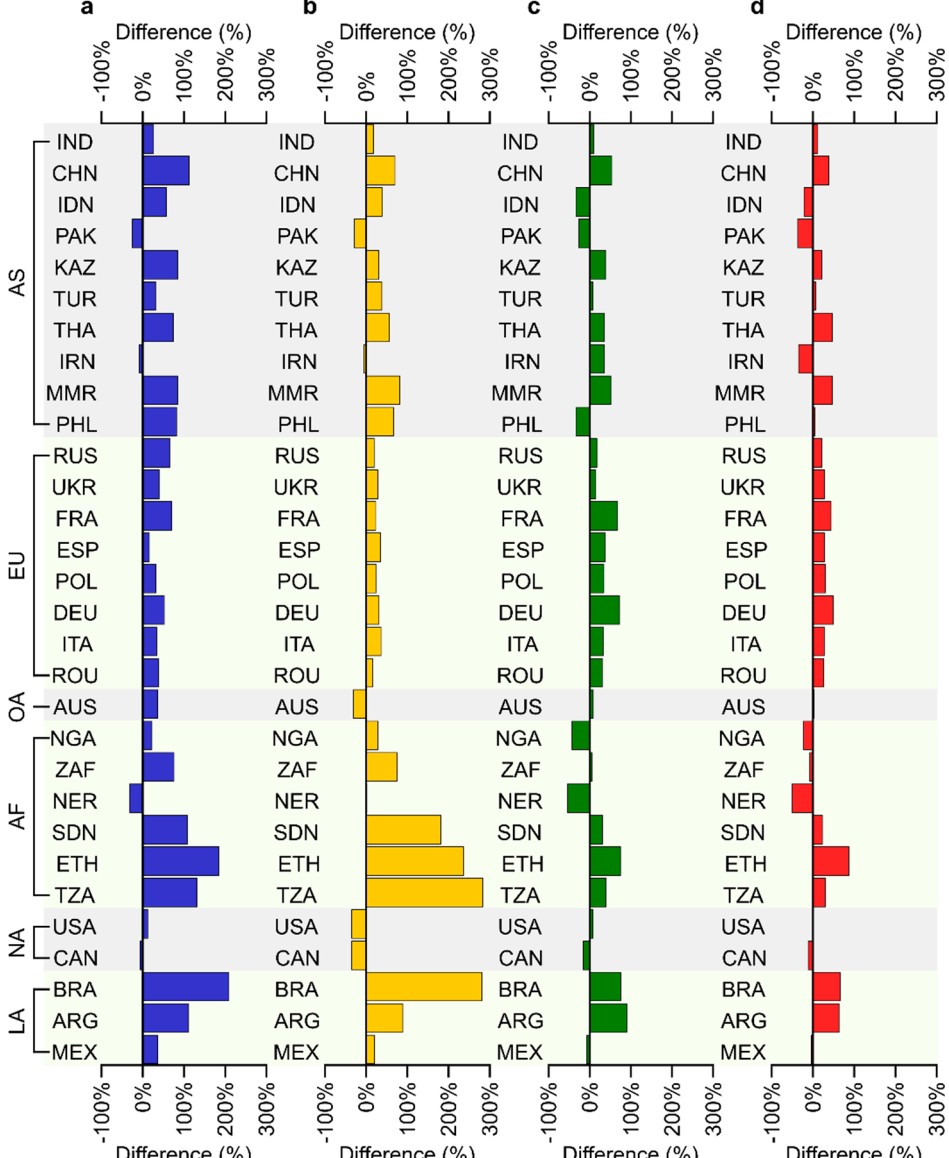

**Figure 7.** Relative differences in national cropland areas (AREA$_{Frac}$ - AREA$_{FAO}$)/AREA$_{FAO}$) in four non-calibrated fractional datasets. The countries in each continent are assorted in descending order according to the FAOSTAT cropland areas. The x-axis represents the relative difference (unit: %); the y-axis shows the country code (named by ISO), where (**a**) shows the GLC-Consensus results, (**b**) shows the ESA-CCI-LC results, (**c**) shows the GlobeLand30 results and (**d**) shows the synergized cropland dataset developed in this study.

The general performances of the differences of cropland area with the FAO statistics for all non-calibrated fractional datasets on a national scale are shown in Figure 8. The range of relative differences in our dataset is the smallest among these datasets on all continents, especially in Asia, Africa, and Latin America. Although the median of our result is negative (many countries' cropland amounts are slightly less than the FAO statistics) in Africa and in Latin America, it is relatively better than the other datasets with larger cropland areas than the FAO statistics.

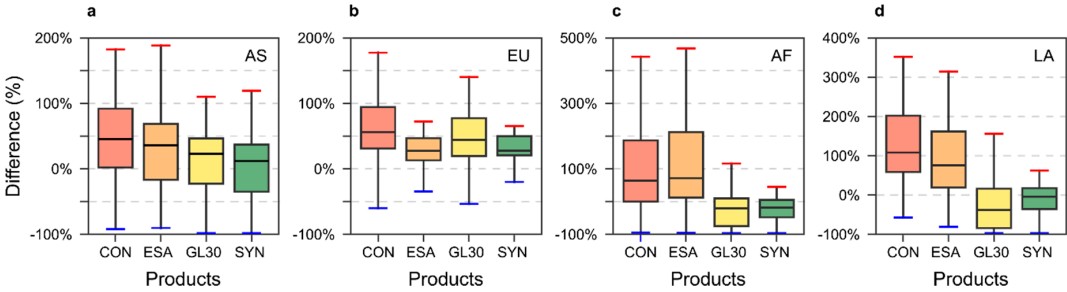

**Figure 8.** Relative differences in the national cropland areas in the non-calibrated fractional datasets. Where (**a**) is Asia, (**b**) is Europe, (**c**) is Africa, and (**d**) is Latin America. The calculation included 48 countries in Asia, 47 countries in Europe, 56 countries in Africa, and 43 countries in Latin America. Abnormal points with very large differences were excluded.

## 4. Discussion

### 4.1. Improvement of Spatial Distribution and Cropland Fraction on Pixel Scale in this Dataset

All synergistic datasets aim to improve the accuracy of spatial distribution and so does our dataset. The accuracy of synergizing datasets has generally improved (even more than 10%) compared with the original satellite-derived products with moderate resolution (1 km × 1 km). The improvement of the synergistic datasets could be attributed to the following two points. Firstly, the cropland pixels in all the original products are allowed to be contained in the new generated synergistic result to indicate the cropland distribution, as some synergistic methods designed (e.g., the method adopted by GLC-Consensus) [47]. That would inevitably increase the number of cropland pixels (where cropland fraction is above 0%) in these synergistic datasets compared to the original Boolean products. Therefore, the probability for the validation samples falling into the cropland pixels will also increase. Secondly, some more accurate regional/national products with finer resolution or interpreted by the domestic institutions or experts have been integrated into the synergistic dataset (e.g., in GLC-Share) [41]. Because the native institutions and experts always have much more professional knowledge in their own regions (e.g., the interpretation task of GLC2000 was separated into different regions and conducted by regional institutions independently) [53].

This study improved the accuracy of spatial distribution mainly by adopting the latter way. By analyzing the accuracy of cropland spatial distribution for each original product, the more accurate ones among the cropland products were identified on the continental/national scale [18] and the products with higher accuracy have been assigned according to the priority order during the synergizing process.

In previous cropland synergizing studies, more concerns were paid to the accuracy of spatial distribution and the consistency with the inventory cropland amount [18,41,46,47]. However, the cropland fraction on the pixel scale, which is of equal importance in synergistic products, is less considered. Comparing with the other synergistic cropland datasets, this study has paid more attention to the rationality of cropland fraction on the pixel scale rather than spatial accuracy and area consistency.

The proportion of cropland pixels with different fractions shows a typical "L" shape, and there is a larger proportion of cropland pixels with higher fraction in our 1 km synergistic fractional dataset. It is similar to the two up-scaling datasets derived from the original products with a high spatial resolution. Considering the common knowledge of both the natural environment and cropland cultivation [56,57], we claim that the "L" shaped curve in the histogram is a reasonable distribution of different cropland fraction based on pixel scale around the world. To meet the enormous food demand caused by large population and increasing needs for material living standards, most parts of the land with favorable agricultural conditions had already been cultivated in the modern world. For example, some regions (e.g., the North China Plain) with a large population density and long agricultural history has developed intensive farming systems; some other regions with better reclamation conditions (e.g., the Great Plain of the U.S. and East European Plain) have been intensively cultivated due to advanced

agricultural machinery and irrigation techniques [58,59]. In addition to a large proportion of pixels with high cropland fraction, a few low-intensity cropland pixels are scattered in the regions with relative poor cultivation conditions around the world. Therefore, most of the pixels in the cropland dataset at a 1 km scale should have a high cropland fraction.

However, the characteristic that a large proportion of cropland pixel with a higher fraction was not well reflected or even missed in most of the previous synergistic products [18,41,47]. That means the proportion of cropland pixels with a higher fraction had been greatly underestimated. Among most of the previous cropland synergizing studies, the cropland fraction on the pixel scale could mainly be estimated from the land cover classes defined by the fractional range in the original Boolean products with moderate resolution (1 km × 1 km), especially for those synergizing process without or with less high-resolution products [18,41,47]. Generally, the cropland fraction defined in those Boolean products is a wide range rather than an exact fractional value [25]. The wide fractional range represented by each class has to be converted to a definite value by taking the mean value of the whole interval before the synergizing process [18,47]. For instance, the cropland class usually ranges from 60% to 100% in 1 km Boolean products, whose cropland fraction is assigned with 80% for synergizing. However, the value in most pixels among this range is more than 90% according to the up-scaling results derived from the high-resolution products (Figure 5b). It is undoubted that using the mean value of the range as the cropland fraction would not only cause obvious underestimation of the cropland fraction on the pixels with higher cropland fractions, but also subjectively homogenize the cropland pixels with various fractions relative to the real ones. In addition, some studies adopted the mean value of different products as the synergized pixel value, rather than selecting the optimal original products. Such a process could further make the cropland fraction diminished on the pixel scale obviously. Finally, the Boolean products with 1 km resolution could not identify the cropland with low fraction, because the pixels with vegetation covers less than 10% or 15% will be defined as a bare land class rather than mixed cropland classes according to the products' land cover classification systems and their detailed definitions [25,27,28,55].

In this study, we made full use of two products with relative higher spatial resolution. During the synergizing process, one optimal product's value was selected as the cropland fraction on the pixel scale of the synergizing result. Firstly, to reduce the uncertainty of estimated cropland fraction derived from the low-resolution Boolean products, the more reliable cropland fraction had been acquired by the up-scaling process. So the reliability of cropland fraction on pixel scale has greatly improved. Secondly, a fine consistency analyzation method of multi-sets' cropland fraction had been developed to find the optimal products for synergizing the assignment of the cropland fraction on the pixel scale, instead of the simple combination of multi-sets' cropland fraction done in most of the previous studies.

Although the new synergistic method is designed to improve the reliability of spatial distribution and cropland fraction on pixel scale, there are still some uncertainties when applied to some regions where the spatial distribution of cropland among different original products has an extremely large discrepancy. Because this method mainly depends on the consistency analysis of cropland intensity ranges among these original products, this discrepancy presents a challenge. The spatial distribution of cropland with extremely large discrepancy among different original products means that the most cropland with a higher fraction in one product is located at the pixels with lower cropland fraction in other products. According to consistency analysis, these cropland pixels were assigned with relative lower fraction during the synergistic process and the cropland area in these regions are also diminished obviously, e.g., the cropland area of Iran is greatly underestimated in our synergistic dataset because of the large discrepancies of cropland spatial distribution among those original products (Figure 7). Moreover, for the region with lower cropland fraction (usually with a relatively harsh agricultural environment, e.g., Finland in northern Europe), it is always hard to be identified as cropland pixels in most of satellite-derived products with moderated resolutions, while it could be well identified in high-resolution products (e.g., GlobeLand30). Thus, the synergistic cropland pixels in these regions are

also easy to be assigned with a lower fraction or even to be identified as non-cropland pixels by our method, because most of datasets used in this study have moderate resolutions.

*4.2. The Uncertainty in Accuracy Assessment of Fractional Dataset Datasets*

Although the spatial distribution accuracy of the new synergistic dataset is lower than the GLC-Consensus's and GlobeLand30's results (Table 3), there is uncertainty in the accuracy assessment of fractional datasets when they are evaluated by validation samples, although such validation method is usually suitable for satellite-interpreted Boolean products. The accuracy of cropland distribution is represented by the proportion of the cropland samples located in the cropland classes that is how many of the cropland pixels were interpreted accurately. When making an accuracy assessment of fractional datasets, the prerequisite is to define how many cropland percentages are dominant in the per-pixel scale could be identified as a cropland pixel. In this study, the pixel with a cropland fraction of more than 10% is defined as a cropland pixel in order to make the validation results of fractional type comparable to the Boolean type products. However, when the samples fall in pixels with a low cropland proportion (<10% but not 0%), it is hard to say that the samples are located in a pixel with no croplands.

The accuracy of fractional datasets evaluated by the validated samples cannot be regarded as an objective reference for assessing the qualities among several datasets. That is, the product just with the highest accuracy among these datasets might not be the best one. For example, the accuracy of GLC-Consensus exceeds 95% (Table 2), but the number of cropland pixels of this product is approximately one times higher than that in most other datasets (Figure 5a). There exist a large number of unreasonable distributions, such as in the Tibetan Plateau of China [47]. Because the product is generated by the synergistic method that directly combines all cropland pixels from original datasets [47] and a large number of cropland pixels in the new synergizing dataset, it would greatly increase the probability of the validation samples falling within the cropland extent.

To avoid the above-mentioned problems, the pixel assignment during the synergizing process in this study was only selected from one reliable product's based on the consistency analysis of reclamation intensity among multi-sets of products. So, comparing with the geographical knowledge of cropland distribution around the world, the total amount of cropland pixels and their spatial distribution in our new dataset are relatively more reliable than the original satellite-derived products. Therefore, when executing the accuracy evaluation for fractional datasets according to the validation samples, it is very necessary to fully consider the relative rationality of the total amount of cropland pixels. Although the accuracy of our synergistic dataset evaluated by validation samples is not the highest compared with the datasets referred, the uncertainty in this study has been partially reduced, especially compared to the GLC-Consensus.

## 5. Conclusions

This study proposed a cropland synergistic method based on consistency analysis of the cropland reclamation intensity among multi-sets of land cover products and developed a set of modern (around 2000 C.E.) fractional synergistic global cropland datasets (1 km × 1 km). The reliability of spatial distribution and cropland fraction on pixel scale of this new synergistic dataset has improved obviously. The main conclusions are shown as follows:

(1) The accuracy of spatial distribution assessed by validation samples in this new synergistic dataset reaches 87.6%. Besides the high accuracy, the dataset has a moderate amount of cropland pixel comparing with the products used in this study;

(2) The reliability of the cropland fraction on the pixel scale has greatly improved that is the large proportion of cropland pixels is with higher fraction (over 90%) in this dataset. The histogram of pixel numbers with different reclamation intensities exhibits an "L" shape that was not found in previous synergistic fractional cropland datasets. This feature is reasonable because it is consistent with the up-scaling results derived from satellite-derived products with high spatial resolutions and the principle of cultivation;

(3)　The cropland areas in this non-calibrated result are generally closer to that of FAOSTAT on the scales from global to national when compared to other non-calibrated synergistic datasets and original satellite-derived products;

(4)　The reliability of the synergistic result developed by this method might decrease to some degree, especially in the regions where there are huge discrepancies among original multi-sets of datasets.

**Supplementary Materials:** The following are available online at http://www.mdpi.com/2072-4292/11/19/2250/s1.

**Author Contributions:** Conceptualization, X.F., Y.Y., and C.Z.; methodology, C.Z.; formal analysis, C.Z. and H.L.; funding acquisition, X.F. and X.W.; writing—original draft preparation, C.Z.; writing—review and editing, Y.Y., X.F., X.W., and H.L.

**Funding:** This research was funded by The National Key Research and Development Program of China (2017YFA0603304) and National Science Foundation of China (grant no. 41807433).

**Acknowledgments:** The authors greatly acknowledge Xue Zheng, Zhilong Zhao, Peng Liang for their useful comments on the manuscript. The constructive comments and kind suggestions of two anonymous referees significantly improved this contribution.

**Conflicts of Interest:** The authors declare no conflicts of interest.

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
