# Peer review of "Synergistic Modern Global 1 Km Cropland Dataset Derived from Multi-Sets of Land Cover Products"

_remotesensing, doi:10.3390/rs11192250_

Round 1

Reviewer 1 Report

An overview

It was with a great interest that I read this rather original work. When reading, one can  quickly appreciate the authors' good mastery of the question. This is also apparent in the abundant literature that is being exploited. Thus, the theoretical framework is clearly and abundantly  defined with concepts well mastered. The references are well cited and relevant to the study. The methodology is adequate and well presented. The objective of the article is to develop a set of data on cultivated land in synergy with a precise spatial distribution and a reasonable recovery intensity per pixel. This is to provide additional data on global changes and estimates of greenhouse gas emissions. The summary is well presented and clearly shows the different results of the article. The authors combine nearly 853 images of one km² each using satellite data. In terms of results, the authors show a classification by continent according to the fraction of space devoted to agriculture. Naturally, the great Indian, Chinese, European, American and African plains are the main areas of land concentration dedicated to agriculture. It could give way and use in individual country dataset. But there are some weak points underligned in the manuscript itself and repeated here.

2.Weak points

In introduction the authors noted the importance that this study can have in the assessment of global changes and the evaluation of greenhouse gas emissions. Unfortunately, they do not come back to discuss the significance of their results for this phenomenon. This can be done in the discussion section if the authors do not want to include in their findings. In one section of their results, the authors dwell at length on the advantages and disadvantages of their methodology. This poses a problem of organizing ideas. This additional information on the methodology should be included in the methodology section at the beginning of the article. In discussion, the reader expects such evaluation on the findings precisely. The authors do not take into account the different forms of agriculture (seasonal, perennial, and irrigated). Consideration of agroforestry, for example, can make a definite contribution to this article. This may explain the underestimation of agricultural area in Africa compared to FAO data. If possible, it would also be better to see clearly how the authors take into account the period of the year that the satellites data were recorded. For the global level, the authors take into account FAO data to discuss their results, some national data should have been taken into account for comparisons made with countries. Because FAO generally recorded statistical data from countries and compiled it and those data are no spatial data. There are so many we (59 times) and our (37 times). It is adviceable to reduce it

There are so many findings merged together and not allowing the reviewer to distinguish clearly between results and discussion.

Reviewer 2 Report

This is an interesting paper and presents a new approach for detecting cropland. 

In general, the introduction could be proof read a bit more. I noticed some awkward phrasing and sentences that did not follow correct grammar. There were also a lot of passive verbs. I made a few notes below on some of the more noticeable mistakes.

Line 37: "an enormous amount" would be better to give a value

Line 59: It is useful to give an approximation of the cell value from degree to km since they are being compared side by side.

Line 65-66: Unclear. Do you mean between the IGBP and Globeland or between the images used within the datasets (e.g. only GL30)?

Line 72-73: The most common synergistic method usually comprehensively considers…

Line 74: …process adopts the mean value…

Line 75: …or chooses one optimal…

Line 80: remove the comma

Line 94-95: “can completely solve the problem” – this is a very strong statement. Is it true?

Line 99: how accurate?

Line 100: What is reasonable or appropriate?

Line 100: …cropland amount in order to…

Table 1 Lines 7-9: what is “around” under year?

Line 304: I do not see a J-shape. Or is this a backwards J, more like an L?

Line 305: the x-axis is the pixel count, not the intensity?

Line 306-307: The sentence “With the intensities increasing from high or low, the number of pixels gradually increases.” This sentence is hard to read. Isn’t the intensity decreasing from the extremes?

Line 313: An inverted U would look like ∩  not Ↄ, which I think better describes the histogram in its orientation.

Lines discussing FAOSTAT around 370: How do you know these are underestimates? Did you verify the FAOSTAT designation? Were the values estimates or reports? Most of the FAOSTAT data have some designation of where the statistic came from. That is important to know how reliable your check data is.

General comment on discussion. I expected the results to be better explained in terms of uncertainty and reliability. I did not feel like the discussion brought further meaning or understanding of the results or the method. This really needs to be addressed because as I read the results I thought "why" a lot and these were never answered in the discussion.

Line 457-458: See earlier comment about the shape and x-axis

Line 458-459: Review quickly the reasons and support the why.

Line 470: …cultivation or gradual abandonment…

Line 496-504: Why might that be? Especially in the example of Iran?

General conclusion comment. What is the overall conclusions of the accuracy and application of the method? How much more reliable is it than the other methods? What about the data result? The conclusion should not be a summary of the paper.

Round 2

Reviewer 2 Report

Dear Authors,

A good redraft. I have minor comments and a few questions that should be clarified in the text.

Please carefully proof-read this text. There are several minor errors. Although the errors do not change the meaning of the text, it is very annoying to the reader. I've only noted a few. 

Line 51: Such as... this is not a sentence. Rather they are examples to the previous sentence. "...studies, such as ..... (refs)."

Line 59: even more fine --> even finer

Line 61: remove "been"

Line 73: do not begin sentences with and.

Line 116: independent China's classification system? Do you mean China's independent classification system?

Line 280: FAOSATA? Do you mean FAOSTAT?

Line 282: unis?

Line 298: Norway, Sweden, and Finland have suspiciously very little cropland.

Line 315: were shown

Line 386: lowercase "nine"

Line 391-399: how do you know about the under/overestimation? What could be the reason? Has it anything to do with the latitude and satellite collection? I do not know.

Line 409-414: why is there this difference? have you checked the FAOSTAT uncertainty?

Line 438: later way? latter? Please be clearer what you are referring to because it seems you mean the expert input.

Line 446: dataset's? Do you mean datasets?
